E-cigarettes versus nicotine patches for perioperative smoking cessation: a pilot randomized trial

Lee Susan M. susanlee.anesthesia@gmail.com suze.lee@gmail.com 1 2 3 4
Tenney Rachel 1 2
Wallace Arthur W. 1 2
Arjomandi Mehrdad 5 6
1 Department of Anesthesia and Perioperative Care, University of California, San Francisco , San Francisco , CA , United States of America
2 Anesthesia Service, San Francisco Veterans Affairs Medical Center , San Francisco , CA , United States of America
3 Department of Anesthesiology, Pharmacology and Therapeutics, University of British Columbia , Vancouver , British Columbia , Canada
4 Department of Anesthesia, Royal Columbian Hospital , New Westminster , British Columbia , Canada
5 Medical Section, San Francisco VA Medical Center, San Francisco CA , United States of America
6 Department of Medicine, University of California , San Francisco , CA , United States of America
Niaura Raymond
Electronic publication date: 2018 Sep 28
Publication date: 2018
Volume: 6
Electronic Location ID: e5609
Received 2018 Apr 9; Accepted 2018 Aug 19
Copyright: ©2018 Lee et al.
Copyright year: 2018
Copyright holder: Lee et al.
License: This is an open access article distributed under the terms of the Creative Commons Attribution License, which permits unrestricted use, distribution, reproduction and adaptation in any medium and for any purpose provided that it is properly attributed. For attribution, the original author(s), title, publication source (PeerJ) and either DOI or URL of the article must be cited.
License URL: https://creativecommons.org/licenses/by/4.0/

Keywords: e-cigarette, Electronic cigarette, Electronic nicotine delivery device, Smoking cessation, Perioperative, Tobacco cessation, Nicotine replacement therapy, Quit smoking, Pilot study, Randomized controlled trial

Funding: UCSF Department of Anesthesia and Perioperative Care funds UCSF Resource Allocation Program Helen Diller Family Comprehensive Cancer Center developmental funds National Cancer Institute Cancer Center Support Grant P30 CA 82103-16 This work was funded by internal UCSF Department of Anesthesia and Perioperative Care funds (San Francisco, California, United States of America) and the UCSF Resource Allocation Program grant, administered by the Helen Diller Family Comprehensive Cancer Center developmental funds from the National Cancer Institute Cancer Center Support Grant (P30 CA 82103-16). E-cigarettes were purchased from NJOY using these funds. NJOY had no involvement in the design, execution, or analysis of the study. The funders had no role in study design, data collection and analysis, decision to publish, or preparation of the manuscript.

==============================
Introduction

Cigarette smoking by surgical patients is associated with increased complications. E-cigarettes have emerged as a potential smoking cessation tool. We sought to determine the feasibility and acceptability of e-cigarettes, compared to nicotine patch, for perioperative smoking cessation in veterans.

Methods

Preoperative patients were randomized to either the nicotine patch group (n = 10) or the e-cigarette group (n = 20). Both groups were given a free 6-week supply in a tapering dose. All patients received brief counseling, a brochure on perioperative smoking cessation, and referral to the California Smokers’ Helpline. The primary outcome was rate of smoking cessation on day of surgery confirmed by exhaled carbon monoxide. Secondary outcomes included smoking habits, pulmonary function, adverse events, and satisfaction with the products on day of surgery and at 8-weeks follow-up.

Results

Biochemically verified smoking cessation on day of surgery was similar in both groups. Change in forced expiratory volume in one second (FEV1) was 592 ml greater in the e-cigarette group (95% CI [153–1,031] ml, p = 0.01) and change in forced expiratory volume in one second to forced vital capacity ratio (FEV1/FVC ratio) was 40.1% greater in the e-cigarette group (95% CI [18.2%–78.4%], p = 0.04). Satisfaction with the product was similar in both groups.

Discussion

E-cigarettes are a feasible tool for perioperative smoking cessation in veterans with quit rates comparable to nicotine replacement patch. Spirometry appears to be improved 8-weeks after initiating e-cigarettes compared to nicotine patch, possibly due to worse baseline spirometry and more smoking reduction in the e-cigarette group. An adequately powered study is recommended to determine if these results can be duplicated.

Introduction

Cigarette smoking is known to increase the risk of surgical site infections, and pulmonary, cardiovascular and other complications in patients undergoing surgery (Moller et al., 2002; Myles et al., 2002; Sorensen & Jorgensen, 2003; Sorensen, Karlsmark & Gottrup, 2003; Warner, 2005). Cigarette smoking is more prevalent in United States veterans than non-veterans, being reported by 25% of veterans surveyed in 2007, compared to 20% of non-veteran adults (Brown, 2010). Furthermore, veterans may face unique challenges in smoking cessation, given the perception that smoking was a normalized part of military life (Gierisch et al., 2012) and the high rate of coexisting mental health disorders (Duffy et al., 2012). An analysis within the Veterans Affairs (VA) healthcare system showed that pulmonary complications, cardiovascular complications, and surgical site infections were mediators of smoking-associated mortality at 6-months and 1-year after elective surgery (Singh et al., 2013). A systematic review has shown that preoperative smoking cessation therapy improves both short and long-term smoking cessation (Thomsen, Villebro & Moller, 2014). In fact, a surgical encounter with the healthcare system has been described as a “teachable moment” that may provide extra motivation for patients to permanently stop smoking. Despite this information, most anesthesiologists do not routinely offer smoking cessation advice to their patients (Kai et al., 2008; Warner et al., 2004).

There is currently a desperate need for more data regarding the use of e-cigarettes and their role in smoking cessation interventions (Palazzolo, 2013). Despite the fact that the FDA has not approved any e-cigarettes for therapeutic use, e-cigarettes are widely marketed with cessation-related claims (Grana, Benowitz & Glantz, 2014) and may have the potential to bridge the gap between smoking cigarettes and abstaining. The limited evidence we do have is that e-cigarettes are modestly effective, similar to transdermal nicotine replacement, in achieving smoking cessation at 6-months with few significant adverse events (Bullen et al., 2013). Several reviews of e-cigarettes have indicated that while e-cigarettes may be helpful for some smokers in quitting smoking, the current evidence is inconclusive due to low quality (Hartmann-Boyce et al., 2016; Malas et al., 2016). Despite this paucity of data, a survey of 112 preoperative patients showed that 55% had tried e-cigarettes and 71% of those that had tried stated that their reason was to quit smoking. We attempted to determine the acceptability of e-cigarettes as a smoking cessation aid and add to the limited existing data on the safety and efficacy of e-cigarette use in smoking cessation, specifically in the perioperative setting where the risks of continued smoking are great and the motivation to stop is high. The aim of this pilot study was to demonstrate the feasibility of e-cigarettes as a perioperative smoking cessation aid, and to generate preliminary data that might allow for larger studies and improved understanding of the safety and efficacy of e-cigarette use, not only in the perioperative environment, but also more broadly in a public health context.

Methods

Study design and participants

This was a pilot two arm, parallel group, randomized controlled trial, conducted at the San Francisco Veteran’s Affairs Medical Center, which is affiliated with the University of California, San Francisco, studying the use of e-cigarettes (versus nicotine patches) for perioperative smoking cessation. Participants were eligible if they presented to the anesthesia preoperative (APO) clinic for elective surgery 3 or more days before surgery, were current cigarette smokers of more than two cigarettes per day having smoked at least once in the last 7 days, and could provide consent. Participants were excluded if they exclusively used other forms of tobacco (e.g., pipe tobacco) or marijuana only, were pregnant or breast-feeding, had an unstable cardiac condition (e.g., unstable angina, unstable arrhythmia), were currently using smoking cessation pharmacotherapy or were already enrolled in a smoking cessation trial, or currently used e-cigarettes on a daily basis. This study was approved by the UCSF Committee on Human Research (14-15274) and the San Francisco VA Human Research Protection Program and was registered on clinicaltrials.gov (NCT02482233) prior to enrollment of the first patient.

Randomization and masking

APO patients who met inclusion criteria and provided verbal and written consent were randomized to electronic nicotine devices (END) or nicotine patches (NRT). Randomization was computer-generated, with randomly permuted block sizes of 3 or 6, in a 2:1 ratio using the ralloc program (Ryan, 2011) in Stata version 13 (StataCorp LP, College Station, TX, USA). A 2:1 randomization ratio was used because we wanted to gather more experience with feasibility using the newer product (END) in the perioperative population compared to the more thoroughly studied NRT (Lee et al., 2013; Moller et al., 2002; Sorensen, Karlsmark & Gottrup, 2003; Thomsen, Tonnesen & Moller, 2009). Allocation was concealed by consecutively numbered, sealed, opaque envelopes. Due to the nature of the intervention, blinding of subjects was not possible. However, healthcare providers were blinded throughout the perioperative period. Outcome adjudicators were blinded wherever possible, but some participants unintentionally unblinded the investigators while reporting side-effects (e.g., reporting a bad taste with inhalation). Smoking cessation and biochemical outcomes were always assessed prior to inquiring about side-effects to minimize this occurrence.

Control group (NRT)

Patients randomized to the NRT group received a 6-week supply of Nicoderm CQ® patches (5 weeks) and placebo patches (1 week) appropriate to baseline nicotine consumption. Those smoking an average of ten or more cigarettes per day were given the 21 mg/day patch for 3 weeks, the 14 mg/day patch for 1 week, the seven mg/day patch for 1 week, and the 0 mg/day patch for 1 week. Participants who reported smoking an average of less than 10 cigarettes per day at baseline were given the 14 mg/day patch for 3 weeks, the seven mg/day patch for 2 weeks, and the 0 mg/day patch for 1 week. The 0 mg/day patches were clear inert Tegaderm™ (3M, St. Paul, MN, USA) patches cut to the size and shape of the nicotine patch. Participants knew that these patches contained no nicotine. The purpose of the placebo patches was to make the 6th week of the NRT group comparable to the 6th week of the END group, which used no-nicotine e-cigarettes. There is a wide range of NRT products (patch, inhaler, lozenge, gum) commercially available. NRT patches were selected for the control group because prior perioperative smoking cessation studies using NRT have used patches successfully and because they are often first-line therapy in standard clinical use at the VA (Lee et al., 2013; Moller et al., 2002; Sorensen & Jorgensen, 2003; Thomsen, Tonnesen & Moller, 2009). A tapering dose was selected in order to be analogous to the reductions in nicotine concentration in the END group, and because dose-tapering of NRT has been previously shown to be successful in aiding cessation in both perioperative (Lee et al., 2013) and non-perioperative studies (Hartmann-Boyce et al., 2018; Stead et al., 2012).

Intervention group (END)

Those allocated to the END group received a 6-week supply of NJOY e-cigarettes (Scottsdale, AZ, USA) and were instructed to use the Bold (4.5%) e-cigarettes ad libitum for 3 weeks, the Gold (2.4%) e-cigarettes ad libitum for 2 weeks and the Study (0%) e-cigarettes ad libitum for the final week. The number of e-cigarettes issued corresponded to the reported baseline cigarettes smoked per day, calculated assuming one NJOY e-cigarette was equivalent to 10 cigarettes. The NJOY e-cigarette is a disposable first-generation e-cigarette that is available for purchase in shops and online. The e-cigarette consists of a battery, an atomizer, which heats the solution, and e-liquid (nicotine and propylene glycol). A photo of the study product is shown in Fig. S1.

First generation e-cigarettes were selected at the time of study design, as they were widely available and evidence that second-generation e-cigarettes were more satisfying to smokers was not yet commonly known (Dawkins et al., 2015). By the time the study was approved by the REB, it was felt that changing products would cause significant delays to starting the study. Furthermore, we selected a simple product that did not require charging or e-liquid refills because our population was older with more co-existing disease (e.g., osteoarthritis) than the typical vaping population and wanted to choose the easiest product with the fewest parts.

Study procedures

Apart from the study product (NRT or END), which was given to patients at the end of the baseline visit, the NRT and END groups underwent identical study procedures. The study consisted of 3 in-person visits (preoperative baseline, day of surgery, and 8-week follow-up) and two phone-calls (30-day postoperative and 6-month follow-up). At each in-person visit, exhaled carbon monoxide (CO) was measured using the piCO™ Smokerlyzer® (Bedfont Scientific Ltd., Kent, England) and forced expiratory volume in the first second (FEV1) and forced vital capacity (FVC) were measured using the EZOne Spirometer (NDD Medical Technologies, Inc, Andover, MA, USA). A saliva sample was collected for cotinine analysis. At each visit and phone call, smoking status was assessed.

Participants in both groups were asked to refrain from the use of cigarettes and all study products at the end of 6-weeks and return unused products at the 8-week visit. Given the lack of long-term safety data for e-cigarettes, we felt it would be unethical to encourage their long-term use. We instructed the NRT group similarly to maintain comparable instructions between groups.

Recruitment/preoperative baseline visit

After written informed consent, baseline demographics, smoking habits, exhaled CO, FEV1, FVC and saliva sample were obtained. Participants were educated on the use of both products prior to receiving their allocated product, which was prepared by the study pharmacist and placed in a brown paper bag to mask the contents to the investigator. All participants received brief counseling by the research team, a brochure produced by the American Society of Anesthesiologists explaining the benefits of preoperative smoking cessation, and a referral to the California Smokers’ Helpline. The referral was an online form completed by the research team, which would trigger a phone call to the participant by the California Smokers’ Helpline.

Day of surgery

Participants were seen by study personnel pre-operatively on their day of surgery. Smoking status, exhaled CO, FEV1, FVC and saliva sample were obtained. Participants were asked about the occurrence of adverse events or side effects related to use of product. For those whose surgical date was cancelled, a make-up day of surgery visit was scheduled to be as close to the original surgical date as possible.

30-day postoperative phone call

Participants were contacted by phone 30-days post-operatively. If participants could not be reached 30-days post-operatively, subsequent attempts were made until contact was established, to a maximum of 10 attempts. Seven-day point prevalence smoking status was assessed by self-report. Adverse events, side effects, and surgical complications were also assessed.

8-week post randomization visit

Seven-day point prevalence smoking status was assessed and exhaled CO, FEV1, FVC and saliva sample were obtained. Participants were asked about the occurrence of adverse events or side effects related to use of product. After these measures, participants revealed product allocation to study personnel. Study personnel conducted a 30–45 min long qualitative interview to assess product usage, and participants’ attitudes toward both products, the results of which are reported separately. For those that refused an in-person visit, but agreed to telephone interview, exhaled CO, FEV1, FVC and saliva were not obtained. Patients were mailed a $100 check after completion of the 8-week follow-up visit (or telephone interview).

6-month follow-up phone call

Seven-day point prevalence smoking status and use of e-cigarettes was assessed.

Statistical analysis

Power calculation

The targeted sample size was 30 (20 intervention, 10 control), balancing cost against precision, as is conventional for a pilot study. Given the small sample size, between-group differences were not expected to be statistically significant. However, the sample size was intended to provide rough estimates of smoking cessation in each group and provide point estimates and confidence intervals needed for planning a full-scale trial. Regarding participant satisfaction with the product, the sample size did have 80% power to detect an effect size of 1.09, accepting an alpha of 0.05 and assuming the standard deviation of a likert scale (0–7) of satisfaction with the intervention (END or NRT) would be around 1.04.

Outcome measures

Outcome measures were predefined. The primary outcome was smoking cessation for at least 48 h on the day of surgery, as confirmed by exhaled CO ≤10 ppm.

Secondary outcomes were:

(1) Smoking cessation at 30-days postoperatively, 8-week and 6-month follow-up.

(2) Smoking reduction of 50% or more (including cessation) compared to, by self-report on the day of surgery, 30-days postoperatively, 8-week and 6-month follow-up.

(3) Change in FEV1, FVC, and salivary cotinine from baseline to day of surgery and 8-week follow-up.

(4) Satisfaction with product, description of product as helpful, and recommendation of the product to others, as assessed by agreement on a 0-7 likert scale on the day of surgery, 30-days postoperatively, and 8-week follow-up.

(5) Adverse events and 30-day postoperative complications.

Analysis plan

The study was analyzed by intention-to-treat. No adjustments for multiple comparisons were made, because all outcomes were pre-specified and were limited to a relatively small number.

Descriptive statistics were calculated for baseline demographic variables. Categorical outcomes were analyzed using Fisher exact test. Histograms were constructed for continuous outcomes and visually assessed for distribution and analyzed using Student t test if normally distributed; Wilcoxon rank sum test was used for non-normally distributed variables. A two-tailed p value of <0.05 was considered significant. Stata version 13 (StataCorp LP, College Station, TX, USA) was used for all data management and analyses.

Results

Patient enrollment

Between August 2015 and February 2016, 30 patients were recruited into the study. Follow-up for the primary outcome was completed in May 2016 and 6-month follow-up calls were completed in August 2016. During recruitment, 1524 patients were booked in the anesthesia preoperative clinic. Of these, 328 (21.5%) were identified as likely smokers based on electronic chart review and 198 were invited to participate in the study. For details, including reasons for missing data, see Consolidated Standards of Reporting Trials (CONSORT) flow chart in Fig. 1.

Figure 1 Consolidated Standards of Reporting Trials flow chart indicating recruitment, randomization and retention of trial participants.

Of the 35 patients approached for inclusion but found to be ineligible, the reasons for ineligibility included: smoking less than two cigarettes per day (n = 10), already being on smoking cessation pharmacotherapy (n = 9), smoking non-cigarette tobacco only (n = 5), prior adverse reaction to NRT patch (n = 3), already enrolled in smoking cessation program (n = 3), regular use of e-cigarettes (n = 2), surgical date changed (n = 2) and currently experiencing an unstable cardiac condition (n = 1). One patient was found to be ineligible after consent, but prior to randomization. All patients were given the treatment (END or NRT) to which they were randomized. Losses to follow-up were minimal and balanced between groups.

Baseline characteristics

Baseline patient characteristics are presented in Table 1. Patient demographics and types of surgery were well balanced. The END group had a higher degree of smoking disease burden, as indicated by a greater number of cigarettes smoked per day, higher Fagerström score for nicotine dependence, increased salivary cotinine and exhaled carbon monoxide levels, and more obstructive spirometry values. Although not statistically significant (p = 0.37), the END group had more diagnosed COPD (n = 6, 30%) than the NRT group (n = 1, 10%) at baseline.

Table 1 Baseline characteristics.

Characteristic	NRT group (n = 10)	END group (n = 20)	
Demographics			
Male	9 (90.0%)	18 (90.0%)	
Age (years)	53 (10.6)	54 (12.7)	
Height (cm)	179.8 (8.9)	180.7 (7.7)	
Weight (kg)	92 (25.9)	97 (19.7)	
Body mass index (kg/m2)	28.5 (7.9)	29.6 (5.8)	
Race (white)	5 (50.0%)	11 (55.0%)	
Ethnicity (latino)	0 (0%)	2 (10.0%)	
Surgery Details			
General surgery	3 (30.0%)	5 (25.0%)	
Orthopedic surgery	2 (20.0%)	6 (30.0%)	
Neurosurgery	1 (10.0%)	1 (5.0%)	
Vascular surgery	1 (5.0%)	0 (0%)	
Other surgery typea	4 (40.0%)	7 (35.0%)	
Ambulatory surgery	7 (70.0%)	13 (65.0%)	
Days seen prior to scheduled surgery	16.5 (9.5)	11.2 (7.9)	
Comorbidities			
Diabetes	0 (0%)	2 (10.0%)	
Hypertension	3 (30.0%)	7 (35.0%)	
Heart diseaseb	0 (0%)	1 (5.0%)	
COPD	1 (10.0%)	6 (30.0%)	
Smoking Habits			
Cigarettes smoked per day	10.8 (6.6)	15.3 (10.5)	
Number of years smoking	32 (16.4)	32 (15.6)	
Pack-years smoking history	16.7 (12.1)	26.4 (27.0)	
Fagerström score (out of 10)	2.5 (0.85)	3.7 (2.6)	
Laboratory indices			
Salivary cotinine (ng/ml)	130.1 (75.3)	209.6 (110.3)	
Exhaled CO level (ppm)	16.1 (7.7)	21.7 (11.5)	
FEV1 (L)	3.14 (1.35)	2.78 (1.11)	
FVC (L)	3.52 (1.28)	4.03 (1.32)	
FEV1/FVC ratio (%)	105% (81.3%)	68.2% (13.0%)	
Notes.

Values are mean (SD) or n (percentage). Percentages may not add to 100 due to rounding.

NRT nicotine replacement therapy

END electronic nicotine delivery (e-cigarette)

BMI body mass index = (weight (kg)/height2 (m2))

COPD chronic obstructive pulmonary disease

CO carbon monoxide

FEV1 forced expiratory volume in first second

FVC forced vital capacity

a Other surgery includes ophthalmology, urology, otolaryngology, plastics, gynecology and podiatry.

b Heart disease defined as coronary artery disease, congestive heart failure, or arrhythmia.

Smoking cessation outcomes

Smoking cessation outcomes are presented in Table 2. There were no statistically significant differences between smoking cessation or reduction rates between NRT and END groups at any time point. However, there was a trend towards improved outcomes in the END group at the 8 week follow-up visit. Biochemically verified smoking cessation for 2 days preoperatively was achieved in 20% (n = 2) of the NRT group, which was similar to the 15% (n = 3) in the END group (p = 1.0). At 8-week follow-up, no participants in the NRT group had biochemically verified smoking cessation, while the END group had three participants (15%) that achieved 7-day point-prevalence abstinence (p = 0.53). When including both those that quit and those that reduced cigarette consumption by at least 50%, 70% of the END group (n = 14) was able to reduce or quit compared to 40% of the NRT group (n = 4), but this difference was not statistically significant (p = 0.14). The number of cigarettes per day smoked by group at baseline, day of surgery, 30-days postoperatively and 8-week follow-up are represented graphically in Fig. S2. Both NRT and END groups reduced their cigarette consumption over time, with median cigarette consumption decreasing from 12.5 [IQR = 8–20] at baseline to 3 [IQR =0.3–9.5] at 8-week follow-up in the END group (p = 0.0001) and from 7.8 [IQR = 6–20] to 5 [IQR = 3–8] in the NRT group (p = 0.01). There were no statistically significant differences between groups in percentage of smoking reduction at any time point (Table 3).

Table 2 Smoking cessation outcomes.

Outcome	NRT group (n = 10)	END group (n = 20)	Relative risk (95% CI)	p	
Day of surgery					
Smoking cessation (verifieda)	2 (20%)	3 (15%)	0.75 (0.15–3.79)	1.0	
Smoking cessation (self-report)	3 (30%)	4 (20%)	0.67 (0.18–2.42)	0.66	
Smoking reduction (including cessation)b	7 (70%)	13 (65%)	0.93 (0.55–1.56)	1.0	
30-days postoperatively			
Smoking cessation (self-report)	3 (30%)	5 (25%)	0.83 (0.25–2.80)	1.0	
Smoking reduction (including cessation)b	5 (50%)	9 (45%)	0.90 (0.41–1.98)	1.0	
8-weeks after randomization			
Smoking cessation (verifieda)	0 (0%)	3 (15%)	RR = undefinedc	0.53	
Smoking cessation (self-report)	1 (10%)	5 (25%)	2.5 (0.34–18.6)	0.63	
Smoking reduction (including cessation)b	4 (40%)	14 (70%)	1.75 (0.78–3.94)	0.14	
6-month follow-up			
Smoking cessation (self-report)	1 (10%)	5 (25%)	2.5 (0.34–18.6)	0.63	
Smoking reduction (including cessation)b	5 (50%)	6 (30%)	0.62 (0.31–1.24)	0.43	
Notes.

Values are n (percentage). p-values from Fisher’s exact test. Relative risks were END versus NRT.

NRT nicotine replacement therapy

END electronic nicotine delivery (e-cigarette)

Cessation on the day of surgery was determined based on 48-hour point prevalence abstinence. Cessation at all other time points was determined by 7-day point-prevalence abstinence. Smoking reduction includes those that quit.

a Smoking cessation verified by exhaled carbon monoxide 10ppm or less.

b Smoking reduction is defined by reduction of 50% or more cigarettes per day compared to baseline, including smoking cessation. Analysis by intention-to-treat—those lost to follow-up were assumed to have continued smoking.

c Relative risk undefined due to no quitters in the NRT group, risk difference = 15% (95% CI [−6.5%-+ 30.6%]).

Table 3 Laboratory outcomes.

Continuous outcomes (laboratory, spirometry, and percent reduction in cigarettes smoked per day).

Outcome	NRT group	END group	Difference(95% CI of difference)	p	
Day of surgery	(n = 10)	(n = 18a)			
FEV1 (ml) change	−236 (585)	−163 (549)	73 (−383 to +528)	0.75	
FEV1/FVC ratio (%) change	−32.2% (74%)	−1.6% (8.2%)	+30.6% (−5.3% to +66.5%)	0.09	
Cotinine (ng/ml) change	+106 (137)	+19 (119)	−87 (−189 to +14)	0.09	
Exhaled CO (ppm) change	+1.9 (7.2)	−1.7 (10.7)	−3.6 (−11.4 to +4.2)	0.35	
Percentage reduction of cigarettes smoked per day	49% (45%)	59% (37%)	10% (42% to −22%)	0.52	
30-days postoperatively				
Percentage reduction of cigarettes smoked per day	51% (31%)	33% (49%)	−18% (22% to −61%)	0.39	
8-weeks after randomization	(n = 8)	(n = 18)		
FEV1 (ml) change	−300 (497)	+292 (503)	+592 (+153 to +1032)	0.01	
FEV1/FVC ratio (%) change	−38.1% (79.2%)	+2.0% (10.5%)	+40.1% (+1.8% to +78.4%)	0.04	
Cotinine (ng/ml) change	+34 (89)	−48 (103)	−82 (−169 to +5)	0.06	
Exhaled CO (ppm) change	+7.1 (11.0)	−2.1 (12.2)	−9.2 (−19.6 to +1.2)	0.08	
Percentage reduction of cigarettes smoked per day	47% (41%)	64% (31%)	17% (45% to −12%)	0.23	
6 months after randomization					
Percentage reduction of cigarettes smoked per day	56% (31%)	33% (35%)	−23% (8% to −55%)	0.14	
Notes.

All values are reported as changes compared to baseline.

CO carbon monoxide

FEV1 forced expiratory volume in first second

FVC forced vital capacity

Values are mean (standard deviation). p-values from two-sided t-tests.

Percentage reduction of cigarettes smoked per day refers to the reduction in smoking compared to baseline, where cessation would be 100% reduction and reducing from 20 cigarettes per day to 10 cigarettes per day would be a 50% reduction.

a n = 18 for day of surgery FEV1, FEV1/FVC, and cotinine, but n = 19 for exhaled CO because one patient in the END group agreed to do exhaled CO, but refused all other tests.

Table 4 Adverse events and postoperative complications.

Outcome	NRT group	END group	p	
Day of surgery	(n = 10)	(n = 19)		
Number with any adverse event, n (%)	5 (50%)	10 (53%)	1.0	
Number with moderate adverse event, n (%)	0 (0%)	1 (5%)	1.0	
PACU complicationsa	0 (0%)	2 (11%)	0.53	
30-days postoperatively	(n = 10)	(n = 19)		
Number with any adverse event	5 (50%)	7 (37%)	0.69	
Number with moderate adverse event, n (%)	0 (0%)	1 (5%)	1.0	
Postoperative complications (by self-report)b	2 (20%)	5 (26%)	1.0	
Postoperative complications (by chart review)c	6 (60%)	5 (26%)	0.11	
8-weeks after randomization	(n = 9)	(n = 20)			
Number with any adverse event	3 (33%)	10 (50%)	0.45	
Number with moderate adverse events, n (%)	1 (11%)	1 (5%)	0.53	
Notes.

Values are n (percentage). p-values from Fisher’s exact test.

Severity of adverse events classified as mild if self-limited and no intervention required, moderate if it required intervention (e.g., oral analgesic for headache), and severe if it required hospitalization. There were no severe adverse events reported in either group at any time point. No participants experienced intraoperative complications.

a Two participants in the END group experienced PACU complications (non-cardiac chest pain, which resolved in PACU and wheezing, which resolved with albuterol administered in PACU).

b Two participants in the NRT group experienced self-reported postoperative complications within 30-days (both wound complications), while five participants in the END group had postoperative complications (two wound-related, two bladder-related and one respiratory).

c All complications assessed by chart-review were wound-related in both NRT and END groups.

NRT nicotine replacement therapy

END electronic nicotine delivery (e-cigarette)

PACU post-anesthesia care unit

Long-term smoking cessation, as assessed by telephone for 7-day point-prevalence abstinence at 6-months, was achieved by 25% (n = 5) of the END group and 10% (n = 1) of the NRT group, which was not statistically significantly different (p = 0.63).

Spirometry and cotinine outcomes

Spirometry (FEV1 and FEV1/FVC ratios), salivary cotinine, and exhaled CO were analyzed by comparing changes from baseline between NRT and END groups and are presented in Table 3. On the day of surgery, both NRT and END groups experienced reductions in FEV1 and FEV1/FVC ratios of similar magnitude and were not statistically significantly different (p = 0.75 and p = 0.09, respectively). At 8-weeks after randomization, the END group had greater improvement in FEV1, FEV1/FVC ratio, with increases of FEV1 of 292 ml (SD 503ml) and FEV1/FVC ratio of 2% (SD 10.5%), compared to the NRT group, which experienced a decrease in FEV1 of 300ml (SD 497 ml) and a reduction in FEV1/FVC ratio of 38% (SD 79%). These differences were statistically significant for change in FEV1 (p = 0.01) and for change in FEV1/FVC ratio (p = 0.04). Point estimates for cotinine and exhaled CO reductions were greater in the END group on both the day of surgery and 8-week follow-up visits, but no differences were statistically significant between NRT and END groups.

Adverse events and postoperative complications

No participants in either group experienced severe adverse events at any time point. As shown in Table 4, adverse event rates were similar between groups on the day of surgery (50% in the NRT group experienced at least one adverse event compared to 53% in the END group, p = 1.0) and at 8-week follow-up (33% in the NRT group versus 50% in the END group, p = 0.45). Product usage was similar between groups (Table 5).

Table 5 Participant usage and satisfaction.

Question	NRT group	END group	p	
Day of surgery	(n = 10)	(n = 19)		
Used product daily or most days, n (%)	5 (50%)	11 (58%)	0.71	
Agreement (Likert scale 1–7) with				
“The product is helpful for quitting smoking,” median [IQR]	5 [3–7]	6 [4-7]	0.59	
“I was satisfied with the product to help with quitting,” median [IQR]	5 [3–6]	6 [4–6]	0.71	
“I would recommend the product to someone interested in quitting smoking,” median [IQR]	6 [5–7]	6 [6–7]	0.73	
8-weeks after randomization	(n = 9)	(n = 20)		
Used product daily or most daysa, n(%)	6 (67%)	16 (80%)	0.64	
Agreement (Likert scale 1–7) with				
“The product is helpful for quitting smoking,” median [IQR]	5 [3–7]	6 [4–7]	0.79	
“I was satisfied with the product to help with quitting,” median [IQR]	5 [3–6]	5.5 [2.5–7]	0.67	
“I would recommend the product to someone interested in quitting smoking,” median [IQR]	7 [6–7]	6 [5–7]	0.46	
Notes.

p-value from Fisher’s exact test for product usage and Wilcoxon ranksum test for all other values. Likert scale was used for agreement (1 = strongly disagree, 2 = disagree, 3 = disagree somewhat, 4 = neither agree nor disagree, 5 = agree somewhat, 6 = agree, 7 = strongly agree).

NRT nicotine replacement therapy

END electronic nicotine delivery (e-cigarette). IQR = interquartile range

a Asked about use while supplies lasted (e.g., considered the participant to have used the product daily or most days if they used the product until they ran out).

No participants in either group experienced intraoperative complications. The rate of postoperative complications was similar in both groups (60% in the NRT group and 26% in the END group, p = 0.11).

Common adverse events related to both NRT and END included headache, nausea, cough, and throat irritation, as shown in Table 6. There were no statistically significant differences between event rates in each group.

Table 6 Specific adverse events.

Adverse event	NRT group (n = 10)	END group (n = 20)	p	
Headache	4 (40%)	4 (20%)	0.38	
Nausea	1 (10%)	5 (25%)	0.63	
Dry cough (persistent)	0 (0%)	2 (10%)	0.54	
Dry cough (intermittent)	1 (10%)	6 (30%)	0.37	
Palpitations	2 (20%)	0 (0%)	0.10	
Throat irritation	3 (30%)	5 (25%)	1.0	
Skin irritation	3 (30%)	2 (10%)	0.30	
Other	6 (60%)	7 (35%)	0.26	
Notes.

Values are n (percentage). p-values from Fisher’s exact test. Events were considered to have occurred if patient reported the symptom at any time point assessed (day of surgery, 30-days postoperatively and 8-weeks follow-up). Patients were also given a phone number to call if they experienced side-effects; there were no calls. No participants reported hospitalization unrelated to surgery. No participants reported pneumonia. Other adverse events in the NRT group included: irritable mood, patch not sticking properly, increased cravings, jitteriness, diarrhea, dry mouth, anxiety, sleepiness. Other adverse events in the END group included: slight wheezing, productive cough, choking sensation, poor appetite, burning sensation, burned lip.

NRT nicotine replacement therapy

END electronic nicotine delivery (e-cigarette)

Smokers’ helpline usage

The California Smokers’ Helpline indicated that contact was made with just over half of the participants. Use of services was similar in the END and NRT groups, as shown in Table 7.

Table 7 California Smokers’ Helpline Services.

Service	NRT group (n = 10)	END group (n = 20)	p	
Referral receiveda	8 (80%)	19 (95%)	0.25	
No contact made	3 (30%)	8 (40%)	0.70	
Reached	5 (50%)	11 (55%)	1.0	
Received counseling	0 (30%)	3 (15%)	0.53	
Received materials	2 (20%)	1 (5%)	0.25	
Refused service	3 (30%)	7 (35%)	1.0	
Notes.

Values are n (percentage). p-values from Fisher’s exact test.

a All participants had a web-based referral confirmed by the study team. However, the California Smokers’ Helpline did not locate the referral for two patients in the NRT group and one patient in the END group.

Satisfaction outcomes

The acceptability of nicotine patches and e-cigarettes were assessed on the day of surgery and 8-weeks after randomization. Regular (daily or most days) use of the product was not statistically significantly different between NRT and END groups (p = 0.71), with about half reporting regular usage on the day of surgery. Usage increased by 8-week follow-up, particularly in the END group, which reported 80% (n = 16) regular use compared with 67% (n = 6) in the NRT group, although the difference was not statistically significant (p = 0.64).

Satisfaction was also similar at both time points between NRT and END groups, with both groups being at least somewhat satisfied with the products they were given, as shown in Table 5. More in-depth explorations of how each product was used, patient attitudes towards smoking cessation in relation to the assigned products, and satisfaction with the products were conducted in qualitative interviews at 8-weeks follow-up and will be reported separately.

Feasibility of study processes

As a pilot study, feasibility of study processes were also analyzed. As with prior perioperative smoking cessation studies (Lee et al., 2013), potential research participants were successfully identified by chart review in the preoperative period, although similarly, more than half of smokers chose not to participate in the research study despite meeting eligibility criteria (Fig. 1). Recruitment may have been further reduced in this trial compared to prior perioperative smoking cessation studies due to the use of non-standard treatments (ENDs) compared to other perioperative studies that used NRT or varenicline (Moller et al., 2002; Thomsen, Tonnesen & Moller, 2009; Wong et al., 2017). Blinding of research staff was successful, although it was not feasible to have participants blinded given the nature of the treatment. Follow-up was nearly complete, with losses to follow-up for the primary outcome occurring only due to cancelled surgeries. Future studies may wish to select a fixed date follow-up (such as 8-weeks post-randomization) for the primary outcome, regardless of surgical date, to maintain consistency between participants and reduce losses due to cancelled and postponed surgeries.

Discussion

In this pilot randomized trial of END versus NRT initiated in the preoperative period, we found that e-cigarettes were a feasible and acceptable intervention to veterans around the time of surgery and had similar smoking cessation rates compared to transdermal nicotine replacement. Given the current need for more controlled data with respect to the use of e-cigarettes for smoking cessation (Hartmann-Boyce et al., 2016) (a practice already popular amongst preoperative patients (Kadimpati, Nolan & Warner, 2015)), this trial has demonstrated important groundwork for future studies.

Our findings are consistent with those of the largest randomized controlled trial of e-cigarettes versus NRT to date (Bullen et al., 2013) and several recent systematic reviews (Hartmann-Boyce et al., 2016; Malas et al., 2016), which also found that e-cigarettes were modestly effective in helping smokers quit compared to NRT. Our protocol differed in its shorter duration of therapy (6-weeks) and tapering nicotine dose in both NRT and END groups, suggesting that a shorter duration of therapy may still be effective in assisting patients in quitting smoking. Similar to prior e-cigarette studies (Hartmann-Boyce et al., 2016), none of our participants experienced serious adverse events, although mild and moderate headache, nausea, cough and throat irritation were common. Quit rates in this study (30% in NRT group and 25% in the END group at 30-days postoperatively) were similar to those found in another pragmatic perioperative smoking cessation study using NRT (Lee et al., 2013) (quit rate at 30-days postoperatively in the NRT group was 29% versus usual care 11%), indicating that both e-cigarettes and NRT are probably more effective than no intervention and can be useful tools to assist patients in quitting smoking at a time when they are highly motivated to quit due to impending or recent surgery.

The differential in exposure time to END versus NRT products prior to the day of surgery (average of 16 versus 11 days) may have introduced some difficulty in interpreting the primary outcome. Future studies may choose a fixed time interval post-randomization (similar to our 8-week follow-up visit) rather than the day of surgery as a primary outcome to standardize this variable between groups.

The improvements in spirometry, for example the increased FEV1 and FEV1/FVC in the END group may indicate that e-cigarette vapor is less harmful than continued cigarette smoking, a claim that is often marketed (Grana & Ling, 2014) despite some controversy in the literature (Konstantinos & Riccardo, 2014; Palazzolo, 2013). These results are especially impressive in light of the fact that baseline characteristics show that the END group was imbalanced towards heavier, more dependent smokers with worse baseline spirometry values. The results could also be explained by a higher baseline prevalence of COPD in the END group in addition to the possible increased smoking reduction and cessation in the END group. Given the small sample size and effort-dependence of spirometry, further studies would be needed to verify the consistency of the spirometry findings.

As is typical of pilot studies, our study was limited by small sample size, such that we were underpowered to detect all but the largest of differences between groups. Nevertheless, we were able to demonstrate the acceptability of e-cigarettes for perioperative smoking cessation, as evidenced by similar satisfaction scores between END and NRT groups, with most patients in the END group indicating that the product was helpful in their quit attempt and that they would recommend the product.

Participants were given a set 6-week supply based on baseline cigarettes per day of either END (per manufacturer’s recommendation) or NRT (daily use) upon enrollment to the study. NRT patients were directed to use the patches daily, while END patients were asked to use the e-cigarette devices ad libitum, with the further instruction that if they ran out of a particular strength of product, they should move on to the next product. A limitation to the study may be that the set amount of product given to the END group may not have been truly ad libitum due to the limited supply. Nevertheless, most participants (15/20 (75%) of the END group) had leftover e-cigarettes and therefore did achieve true ad libitum use.

Another limitation to this study was the inconsistent use of behavioral support. While most referrals were appropriately received by the California Smokers’ Helpline, a large proportion were unable to connect with counselling support. It is likely that with better adherence to telephone counselling, quit rates in both groups may have been higher. Future studies may benefit from tighter control of access and use of smoking cessation counselling, including face-to-face counselling and improved follow-up counselling, which might improve smoking cessation rates in both arms, as intensive smoking cessation counselling has previously been shown to be effective in the perioperative period (Thomsen, Villebro & Moller, 2014).

Conclusions

E-cigarettes were found to be a feasible and acceptable aid for perioperative smoking cessation with quit rates comparable to nicotine replacement patch. Spirometry may be improved with e-cigarette use. A large, adequately powered study is recommended to determine if the results from this pilot study can be duplicated.

Supplemental Information

Figure S1 NJOY electronic nicotine delivery device provided to participants in the END group

Each participant in the END group was issued 6-weeks of NJOY e-cigarettes, pictured above. The number of e-cigarettes issued corresponded to their baseline smoking, with 1 NJOY e-cigarette equivalent to 10 cigarettes per day. Participants were given 3 weeks of Bold (4.5%), 2 weeks of Gold (2.4%) and 1 week of Study (0%) e-cigarettes and instructed to use them ad libitum in lieu of their usual cigarettes. END=electronic nicotine delivery.

This photograph was taken by the author Susan M. Lee.

Click here for additional data file.

Figure S2 Smoking by END and NRT groups at enrollment, day of surgery, 8-week visit, and 6-month phone call

Boxplots indicating distribution of cigarettes smoked per day at baseline (Preoperative Clinic visit), day of surgery, 8-week, and 6-month follow-up visits. There were no statistically significant differences between groups END and NRT at any time points. However, both END and NRT groups had significantly lower cigarette consumption at each follow-up visit compared to baseline, as compared using Wilcoxon Signed-Rank tests (p = 0.0004, 0.0001, and 0.0004 for END group comparing baseline visit to day of surgery, 8-week and 6-month follow-up visits respectively; p = 0.02, 0.01, and 0.008 for NRT group comparing baseline visit to day of surgery, 8-week and 6-month follow-up visits respectively). END, electronic nicotine delivery; NRT, nicotine replacement therapy.

Click here for additional data file.

Supplemental Information 1 Study Protocol

Click here for additional data file.

Supplemental Information 2 CONSORT checklist

Click here for additional data file.

The authors wish to acknowledge research assistants Sarah Dalton and Wendy Ching from the Arjomandi Lab and Rosalind Franklin University medical student Christopher Sirivoranankul for their assistance in carrying out this study. The cooperation of the Benowitz Lab is also appreciated for performing salivary cotinine analyses. This work is attributed to the San Francisco Veterans Affairs Medical Center Anesthesiology Service, which is affiliated with the Department of Anesthesia and Perioperative Care and Department of Medicine, University of California, San Francisco.

Additional Information and Declarations

Competing Interests

Author Contributions

Human Ethics

Clinical Trial Ethics

Data Availability

Clinical Trial Registration

The authors declare there are no competing interests.

Susan M. Lee, Rachel Tenney and Arthur W. Wallace conceived and designed the experiments, performed the experiments, analyzed the data, contributed reagents/materials/analysis tools, prepared figures and/or tables, authored or reviewed drafts of the paper, approved the final draft.

Mehrdad Arjomandi conceived and designed the experiments, performed the experiments, contributed reagents/materials/analysis tools, approved the final draft.

The following information was supplied relating to ethical approvals (i.e., approving body and any reference numbers):

This study was approved by the UCSF Committee on Human Research (14-15274) and the San Francisco VA Human Research Protection Program.

The following information was supplied relating to ethical approvals (i.e., approving body and any reference numbers):

The University of California, San Francisco (UCSF) and San Francisco VA Human Research Protection granted ethical approval to carry out the study.

The following information was supplied regarding data availability:

For ethical reasons of participant confidentiality, in accordance with Veterans Affairs data procedures, an anonymized data set has been provided for reviewers, but cannot be published owing to the small size of the study, known location and years of the study which may make elements of the data identifiable.

The following information was supplied regarding Clinical Trial registration:

ClinicalTrials.gov: NCT02482233.

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
