# Peer review of "E-cigarettes versus nicotine patches for perioperative smoking cessation: a pilot randomized trial"

_PeerJ, doi:10.7717/peerj.5609_

## Round 0.1 · original submission · Major Revisions

Please be sure to address all of the reviewers' comments in your response.

Reviewer 1 ·

Basic reporting

The paper reports on a small pilot/feasibility study conducted 2 years ago. It is well-written in clear English to high standards in a professional style. The authors commendably and correctly adhere to reporting guidelines (CONSORT). The references are appropriate and relevant; there is one figure (CONSORT trial flow chart) and 7 tables; raw data is provided.

Experimental design

This is a RCT pilot study. It has been well conducted as a pilot trial, with the study Methods reported in line with CONSORT guidelines, with each of the key CONSORT items addressed specifically.

However, the aims are less than clear - as a pilot trial, it was not powered to detect an efficacy outcome: the concern with estimating a likely effect size with which to help calculate the sample size for a future adequately powered efficacy and safety trial is to the fore. However, it is stated that "The study was designed to test the feasibility and acceptability of the use of e- cigarettes (versus nicotine patch) as a smoking cessation aid in the perioperative period." but many unanswered questions that relate to assessing the feasibility of key study processes (recruitment, blinding, follow up and loss to follow up). are now specifically addressed. And then "We attempted to determine the acceptability of e-cigarettes as a smoking cessation aid and add to the limited existing data on the safety and efficacy of e-cigarette use in smoking cessation, specifically in the perioperative setting where the risks of continued smoking are great and the motivation to stop is high." The acceptability of the study treatments was assessed, but the a one-dimensional scale with a non-validated question about product satisfaction seems weak methodologically.

From a feasibility perspective, it would be helpful to have sought the perspectives of participants and clinicians via complementary qualitative work, such as through focus groups or key informant interviews.

A major design feature that is not clear in the article is the time period exposed to the products pre-operatively - it is unclear how much time each group had using the different study products; analysis of the raw data shows this varies quite widely. This should be reported.

There are other areas that lack clarity and require explanation:
- It is unclear why there was a 2:1 ratio used to randomised (ENDS: patches) and why were patches chosen as the NRT product , rather than a short acting NRT such as the inhalator, which is more analogous to ENDS.
- It is unclear why tapering was used in the dose of nicotine.
- Why was a 1st generation ENDS used when the ability these devices have to deliver nicotine reliably at doses likely to substitute for conventional cigarettes and mitigate withdrawal syndrome symptoms is known to be limited?
- Why were both eCO and cotinine measured if this was a feasibility study?
- What were Lung Function tests done if this was simply a pilot or feasibility study?

The pragmatic approach to accessing counselling support is a design strength in a pragmatic trial but in this situation should have been explored further in the analysis as a key feasibility issue to help decide what to do in the future trial.

Validity of the findings

Validity is limited by the varying time periods of participant exposure to the products prior to surgery.
Could the FEV1 differences be due to the higher prevalence of COPD (30%) in the ENDS group?

Additional comments

Some minor corrections:

Lines 105-106 are redundant
Reference 15 is to a trial that involved a now obsolete e-cigarette. ENDS have improved since in terms of nicotine delivery and reliability so one would expect efficacy for cessation to have improved.
Reference line 341 spelling should be 'Benowitz'
Table 1 use the same font throughout
Table 4 p value 'from'

Reviewer 2 ·

Basic reporting

No comment

Experimental design

In general the experimental design is OK. I have some specific comments that i will submit below and ask for some clarifications and additional discussion by the authors.

Validity of the findings

No comment

Additional comments

This is an interesting research subject. I agree with the authors that it should be further studied. I fully understand the limitations of the small sample size which makes statistical comparisons difficult. These limitations are adequately addressed by the authors.
I have some comments, mentioned below, that the authors need to comment on in the manuscript.

The authors should provide more detailed explanation on why they chose to taper the dose, gradually reducing the levels of nicotine in both groups.

Abstract, Results section, line 55: “30 patients were recruited”
There is no need for this sentence since you already mention in the previous section that you randomized 10 people in the patch and 20 in the e-cigarette group.

Lines 141 and 143-145.
While the authors mention that participants were asked to use the e-cigarettes ad libitum, they seem to have provided limited nr of devices, based on the nr of cigarettes the participants were smoking. I don’t think this can be considered ad libitum use. Also, although this may have been done to follow the same procedure as with NRTs, I don’t think this is appropriate for an e-cigarette trial because it deprives participants from using them as they prefer and based on self needs. Unlike medications, e-cigarettes do not have a recommended dose. Additionally, factors such as depth of inhalation, and puffing patterns may affect nicotine intake from the e-cigarettes, while such factors are absent when NRTs are provided. Please provide some explanations for these issues and perhaps mention some limitations. I believe a less conventional approach is needed when assessing e-cigarettes in clinical trials, with more freedom of choice for flavors and nicotine strengths, as well as true ad libitum use. Perhaps the authors could make some brief statements about these issues in the discussion section.



6-month follow-up.
Were participants instructed to use (if they wanted) or refrain from use of NRTs or e-cigarettes after the 8-week period of providing the products?


Lines 204-206
While smoking cessation and reduction was assessed at 6 months, nothing is mentioned in the results section. There is some information in tables, but I suggest to add a sentence about the 6 months data in the text too.


Line 255.
The authors present the difference in cigarette consumption separately in e-cigarette users and NRT users. I think it would be interesting to compare the % change in cigarette consumption in the two groups and whether there was a statistically significant difference in the % change between groups.


Lines 317-320.
The changes in FEV1 and FEV1/FVC were much more pronounced in e-cigarette users. The authors should try to explain this. For example, e-cigarette users had worse baseline spirometry values indicating more adverse respiratory effects from (the heavier) smoking. Thus, they had more prospects for improvement, while no improvement can be expected in participants who already have good respiratory function at baseline. I noticed there were many more patients with COPD in the e-cigarette group. This would perfectly explain the improvement in respiratory function observed in this group, based on my previous statement. I think this difference in COPD prevalence should be specifically mentioned in the text, and also present whether there was any statistical significance. Also, the higher rates of smoking cessation and reduction in the e-cigarette group, although not statistically significant due to the small sample size, could be another reason for these findings. I don’t think e-cigarettes have any direct beneficial effect on respiratory function, it is smoking cessation and reduction that probably contributed to these findings.

Table 2. I think it would be useful to add rows evaluating the combined smoking cessation/50% reduction rates in both groups and the statistical comparisons.

---

## Round 0.2 · accepted · Accept

Thank you for your thorough response to the queries raised by the reviewers,

Reviewer 1 ·

Basic reporting

The revised article is much improved: it is self-contained, with good clarification of previously inadequately explained elements to the study design or implementation. The writing is clear and professional.
The references are adequate. All required components -tables, data, etcetera -have been shared.

Experimental design

The design has now been clarified in the areas identified in the review where there was unclear rationale or insufficient details about what was done.

Validity of the findings

The findings are now situated within a number of limitations that were previously not included. This helps with interpretation of the study results and the implications (as a pilot study) for next steps.